# PPARα Agonist Oral Therapy in Diabetic Retinopathy

**DOI:** 10.3390/biomedicines8100433

**Published:** 2020-10-19

**Authors:** Yohei Tomita, Deokho Lee, Kazuo Tsubota, Toshihide Kurihara

**Affiliations:** 1Laboratory of Photobiology, Keio University School of Medicine, Tokyo 160-8582, Japan; y.tomita@keio.jp (Y.T.); deokholee@keio.jp (D.L.); 2Department of Ophthalmology, Keio University School of Medicine, Tokyo 160-8582, Japan; 3Department of Ophthalmology, Boston Children’s Hospital, Harvard Medical School, Boston, MA 02115, USA; 4Tsubota Laboratory, Inc., Tokyo 160-0016, Japan

**Keywords:** diabetes retinopathy (DR), antioxidants, advanced glycation end products (AGEs), vascular adhesion protein-1 (VAP-1), microglial activation inhibitor, fenofibrate, selective peroxisome proliferator-activated receptor alpha modulator (SPPARMα), fibroblast growth factor 21 (FGF21), pemafibrate

## Abstract

Diabetic retinopathy (DR) is an eye condition that develops after chronically poorly-managed diabetes, and is presently the main cause for blindness on a global scale. Current treatments for DR such as laser photocoagulation, topical injection of corticosteroids, intravitreal injection of anti-vascular endothelial growth factor (VEGF) agents and vitreoretinal surgery are only applicable at the late stages of DR and there are possibilities of significant adverse effects. Moreover, the forms of treatment available for DR are highly invasive to the eyes. Safer and more effective pharmacological treatments are required for DR treatment, in particular at an early stage. In this review, we cover recently investigated promising oral pharmacotherapies, the methods of which are safer, easier to use, patient-friendly and pain-free, in clinical studies. We especially focus on peroxisome proliferator-activator receptor alpha (PPARα) agonists in which experimental evidence suggests PPARα activation may be closely related to the attenuation of vascular damages, including lipid-induced toxicity, inflammation, an excess of free radical generation, endothelial dysfunction and angiogenesis. Furthermore, oral administration of selective peroxisome proliferator-activated receptor alpha modulator (SPPARMα) agonists may induce hepatic fibroblast growth factor 21 expression, indirectly resulting in retinal protection in animal studies. Our review will enable more comprehensive approaches for understanding protective roles of PPARα for the prevention of DR development.

## 1. Introduction

Diabetic retinopathy (DR) is a complication of diabetes that affects the eyes in subjects with type 1 or type 2 diabetes mellitus [1]. DR develops because of a chronic abnormality of glycemic control [2]. In detail, DR progresses from an initial stage where high blood glucose levels damage the microvasculature [3]. Then, microvascular irregularities including hemorrhage, ischemia, and microaneurysms bring retinal neovascularization [3]. Abnormal vasculatures by retinal neovascularization lead to a severe hypoxic condition in the retina of the eye [3]. At the final stage, fibrovascular proliferation resulting in tractional retinal detachment by chronic severe hypoxic conditions causes vision loss [3]. However, recent evidence indicates that the pathogenesis of DR contains far more complex mechanisms that are the involvement of multiple interlinked alterations via impairment of crosstalk between retinal neurons, glial cells, and vasculatures [4,5,6,7]. Emerging studies have demonstrated that neurons and glial cells in the central nervous system act as oxygen sensors and vascular regulators to interact with vascular cells for neurovascular homeostasis [8], and impairment of their crosstalk damages neurovascular homeostasis [8,9]. Although the exact mechanisms have not been completely defined, possible major key factors such as mitochondrial oxidative stress, inflammation, production of advanced glycation end products (AGEs), and activation of the protein kinase C (PKC) pathway have been proposed (along with the traditional concept of retinal neovascularization by hyperglycemia-induced microvascular irregularities) for the comprehensive pathogenesis of DR (Figure 1) [10,11].

The progression of DR development can be promoted by not only the degree of blood glucose level dysregulation but also that of hypertension and hyperlipidemia [12]. As DR is the leading cause of blindness in industrialized countries [13], many researchers have tried to develop treatments for DR. Dramatic advances in the diagnosis and treatments of DR have been made [14]. To date, treatments for DR are laser photocoagulation, topical injection of corticosteroids, intravitreal injection of anti-vascular endothelial growth factor (VEGF) agents, and vitreoretinal surgery [15]. These treatments are applicable only at the late stages of DR and may cause significant side effects [16,17]. Furthermore, the forms of treatment available for DR are highly invasive to the eyes, which is not patient-friendly [16,17]. The deficiency of safer and more effective treatments and lack of accurate management of DR remain unsolved. Therefore, safer and more effective pharmacological treatments are required for DR treatment, in particular at an early stage. In this aspect, new treatments via oral administration have emerged, which are new pharmacological treatments for DR [18]. Oral therapies include the beneficial features of safety, good patient compliance, ease of ingestion, and pain avoidance [19]. Additionally, experimental and clinical evidence has unraveled beneficial effects of peroxisome proliferator-activator receptor alpha (PPARα) agonists via oral administration on the prevention of DR development [20].

PPARα is from a nuclear receptor PPAR family (PPARα, PPARδ, and PPARγ), which regulates the expression of several genes affecting lipid and carbohydrate metabolism [21]. PPARα is named so based on its ability to be activated by peroxisome proliferator chemicals, and it is the first member to be cloned among the PPAR isotypes [22]. PPARα is expressed in various types of cells in the skeletal muscle, heart, liver, brown adipose, kidney, intestinal mucosa, adrenal gland, eye, and most cell types present in the vasculature including endothelial cells, smooth muscle cells, monocytes, and macrophages [23,24,25,26,27,28,29,30]. PPARα activation was found to increase circulating levels of high-density lipoprotein cholesterol and decrease serum levels of triglycerides, free fatty acids and apolipoprotein, which improves the overall serum lipid profile and finally exerts positive effects on inflammation and insulin resistance [31,32,33]. Increasing evidence suggests that PPARα activation can be a strong therapeutic target for various types of diseases such as cardiovascular diseases [34], dyslipidemia [35], and diabetes and its complications including DR [31]. However, its molecular mechanisms are far from being elucidated. In this paper, we review the therapeutic effects of PPARα agonists as a promising approach for the treatment of DR and shortly cover other on-going oral administration drugs.

## 2. A Comparison of PPARα Agonists and Other Oral Therapies

As diabetes is the most well-known metabolic disorder, there have been lots of approaches to target various molecular pathways to manage DR [36]. Recently investigated oral pharmacotherapies in clinical trials for DR or diabetic macular edema (DME) are listed in Table 1 (https://clinicaltrials.gov*)* (last updated, 6 September 2020)

### 2.1. Antioxidants

Free radicals are natural by-products of biochemical reactions in our body [37]. Oxidative stress can occur when cells cannot adequately clear up an excess of free radicals [37]. The excess of free radicals is commonly observed in diabetic patients [38,39]. The concept of the application of antioxidants is based on the observation that an increase in oxidative stress associated with abnormal glycemic control can contribute to retinal cell damages leading to DR [40]. As a potent antioxidant, α-lipoic acid, a natural compound found in vegetables and meat, may show a protective effect in DR [41]. The administration of α-lipoic acid reduced retinal cell death by activating AMP-activated protein kinase in diabetic mice [42]. The oral administration of α-lipoic acid in combination with genistein and vitamins exerted a protective effect on retinal cells as detected and analyzed by electroretinography in pre-retinopathy diabetic patients [43]. In addition, the protective effect of α-lipoic acid was under evaluation on the occurrence of DME in subjects with type 2 mild DR (Table 1). Even though visual acuity remained unchanged during the entire trial, the study showed a negative result, suggesting a daily dosage of α-lipoic acid may not prevent the occurrence of DME in diabetic patients [44].

Another antioxidant ubiquinone, coenzyme-Q, is a mobile component of mitochondrial electron transport chain and is essential for mitochondrial energy production [45]. Ubiquinone can recycle and regenerate other antioxidants such as tocopherol, the major forms of vitamin E, and ascorbate, vitamin C [45,46]. In a pre-clinical study, ubiquinone attenuated reactive oxygen species and enhanced antioxidant capacity in endothelial cells by activating endothelial nitric oxide synthase and suppressing inflammatory mediators [47]. In a clinical study, ubiquinone and a combined antioxidant therapy in DR improved mitochondrial homeostasis and diminished energy catabolism (Table 1) [48]. However, the direct protective effect of ubiquinone in DR still needs to be elucidated.

### 2.2. Advanced Glycation End Products (AGEs) Inhibitor

The formation and accumulation of AGEs have been nominated as one of the mechanisms of pathogenesis which may contribute to DR development [49]. AGEs are formed from proteins, lipids, and DNA via various complex pathways such as non-enzymatic glycation and reactions with ascorbate, metabolic intermediates, and reactive dicarbonyl compounds, and their accumulation causes severe protein dysfunction [50,51]. In this regard, inhibition of AGE formation may be a promising target for therapeutic intervention in DR development. AGE inhibitors, especially aminoguanidine, attenuate the microvascular lesion formation in the diabetic retinas [52,53]. In a clinical study, restoration of the retinal vasculature by the oral administration of aminoguanidine was under evaluation in type 1 diabetic patients (Table 1), the results of which are yet to be shared with the public (ClinicalTrials.gov Identifier: NCT02099981).

### 2.3. Vascular Adhesion Protein-1 (VAP-1) Inhibitors

VAP-1 is a dual function molecule which possesses enzymatic activity and adhesive properties for leukocyte recruitment [54]. VAP-1 is exclusively expressed on the cellular surface of vascular endothelial cells as well as smooth muscle cells [54,55]. While VAP-1 is mainly absent on the surface of the cells under normal conditions, VAP-1 can be translocated to the cell surface and facilitate the accumulation of inflammatory cells into injured or damaged tissues in concert with other leukocyte adhesion molecules under inflammatory conditions [56,57]. In the eyes of humans and rodents, VAP-1 is localized in choroidal vessels and retinal endothelial cells [58,59,60]. Previous studies have suggested that VAP-1 could be involved in the molecular mechanisms of uveitis [61], choroidal neovascularization [59], and leukocyte transmigration in the retinas of diabetic mice [60]. These findings indicate that VAP-1 inhibition can attenuate severe inflammation in ocular diseases. In this regard, a protective effect of ASP8232 on DR was under evaluation as a VAP-1 inhibitor in diabetic patients (Table 1). Unfortunately, inhibition of plasma VAP-1 activity by ASP8232 showed no effect on central subfield thickness in patients with center-involved DME [62]. Nonetheless, BI 1467335, another VAP-1 inhibitor, has been under evaluation (Table 1), the results of which are yet to be shared (ClinicalTrials.gov Identifier: NCT03238963).

### 2.4. Microglial Activation Inhibitor

For retinal homeostasis, interactions between retinal neurons, glia, and blood vessels are important (neurovascular couplings) [63,64]. There are three main types of glial cells found in the mammalian retina: astrocytes, Müller cells, and microglia [63]. Several previous studies have indicated that an increase in the Müller cell population and a decrease in the astrocytic population could be seen in the early DR, which may be associated with increased vascular permeability in DR [65,66]. Similarly, microglial activation was observed in the early DR in several in vivo studies [67,68]. Even though the pathogenesis of DR remains unclear, evidence suggests that DR at least contains chronic inflammation [69,70] and the microglia can be considered one of the principle cells sensing the stimuli in diabetic conditions and releasing pro-inflammatory and neurotoxic cytokines [67]. In this regard, minocycline, a microglial activation inhibitor, reduced pro-inflammatory cytokine expressions, and suppressed an apoptosis marker caspase-3 activity in rat diabetic retinas [67]. The safety and efficacy of minocycline as a microglial activation inhibitor was studied in the treatment of DME (Table 1). The study showed that minocycline was associated with improvements in visual function, central macular edema, and vascular leakage, suggesting that microglial inhibition by oral administration of minocycline may be a powerful treatment for DME (ClinicalTrials.gov Identifier: NCT01120899) [71].

### 2.5. Peroxisome Proliferator-Activator Receptor Alpha (PPARα) Agonists

Control of blood glucose, blood pressure, and blood lipids is the major method of medical management to prevent the progression of DR [72]. PPARα is a strong regulator of lipid metabolism in response to fasting and evidence is emerging for a role of PPARα in balancing glucose homeostasis as well [21,73]. Fenofibrate, a well-established PPARα agonist, is used to treat hyperlipidemia by lowering triglyceride levels and increasing high-density lipoprotein cholesterol levels [74]. Based on its therapeutic effects on the modulation of lipid metabolism, fenofibrate has been tested for the prevention of DR [75]. Interestingly, fenofibrate treatment led to inhibition of VEGF and VEGF receptor expressions in human retinal pigment epithelial (RPE) cells under a hypoxic condition [76]. Furthermore, its conditioned medium from human RPE cells with fenofibrate under hypoxia lowered the ability of human umbilical endothelial cells to lead to the formation of new blood vessels [76]. The fenofibrate intervention and event lowering in diabetes (FIELD) study showed that use of fenofibrate could reduce the need for any first laser photocoagulation in subjects with pre-existing retinopathy [77]. In the action to control cardiovascular risk in diabetes (ACCORD) study, even though there was no statistically significant difference between the placebo and fenofibrate-administered groups regarding the percentage of patients with moderate vision loss, a reduction of DR progression was observed in the fenofibrate-administered group [78]. Other therapeutic effects of fenofibrate on the regulation of endothelial progenitor and circulating progenitor cell levels have been under evaluation in DR patients (Table 1), the outcomes of which are yet to be shared with the public (ClinicalTrials.gov Identifier: NCT01927315). Choline fenofibrate (SLV348), a newly developed choline salt of fenofibric acid, is more hydrophilic than fenofibrate, and has been evaluated on the regression of macular edema in the eyes of type 2 diabetic patients (Table 1), the results of which will be shared with the public in due course (ClinicalTrials.gov Identifier: NCT00683176). However, for clinical uses, fenofibrate has been associated with a high risk of renal failure with increased creatinine levels in the blood [79]. This is because fenofibrate is primarily excreted from the kidney [79]. Excreted levels of fenofibrate could decrease in patients with impaired renal function. [79]. Therefore, it is not highly recommended for patients with severe renal diseases to receive a fenofibrate therapy [80].

Pemafibrate is a novel selective PPARα modulator (SPPARMα) and has higher potency and selectivity for the activation of PPARα than fenofibrate [81,82,83,84]. A previous report indicated that pemafibrate reduced triglyceride levels and increased high-density lipoprotein cholesterol levels better than fenofibrate [85]. In addition, pemafibrate showed a smaller possibility of kidney-associated adverse outcomes than fenofibrate [85]. These differences may be explained with structural differences of fenofibrate and pemafibrate (Figure 2A). The structure of pemafibrate contains not only the carboxylic acid group (Group A), which fenofibrate consists in, but also the phenoxy alkyl group (Group B) and 2-aminobenzoxazolic group (Group C) (Figure 2) [86,87,88]. Hydrophobic interactions of Groups B and C with the ligand-binding pocket of PPARα increase ligand/receptor binding affinity and the flexibility of Group B confers the stronger “induced fit” conformation with PPARα (Figure 2A) [89]. Binding of pemafibrate with high affinity can induce specific structural transitions of PPARα and recruitment of specific co-factor complexes [90,91,92]. Finally, pemafibrate could exert on-target effects of PPARα activation while fenofibrate could show not only on-target but also off-target effects, such as deleterious effects on the renal function (Figure 2B) [90,91,92,93].

In summary, PPARα agonists could directly affect the pathological mechanism that damage the retina under disease conditions similar to the other oral therapies. Moreover, PPARα agonists could target the modulation of systematic metabolism through control of blood glucose and lipid levels, unlike the other oral therapies.

## 3. Functions of PPARα in the Eye

Experimental evidence indicates that PPARα is expressed in various tissues in diabetic microvascular diseases—the retina as well as kidney and nerve [29,30]. PPARα has been spotlighted as its expression levels have been shown to be reduced in the retinas with diabetes [94,95]. Decreased PPARα expression has been found to contribute to retinal inflammation and neovascularization, and pharmacological activation of PPARα has been found to exert therapeutic effects against various ocular degenerative disorders [95,96,97]. A previous study showed that more severe retinal acellular capillary formation and pericyte dropout were observed in PPARα^−/−^ mice with diabetes, compared with those in diabetic wild-type mice [96]. Another study demonstrated that retinal neurodegeneration analyzed by electroretinography was exacerbated in PPARα^−/−^ mice with diabetes in comparison with that in diabetic wild-type mice [98]. Multiple proteomics data indicated that several oxidative stress markers such as *Gstm1* (glutathione-s-transferase m1), *Prdx6* (peroxidase 6) and *Txnrd1* (thioredoxin reductase 1) were increased in diabetic retinas and that there were further increases in diabetic PPARα^−/−^ retinas [98]. This implies that oxidative stress may become worsened by PPARα ablation in DR. PPARα activation increased retinal NADH (nicotinamide adenine dinucleotide + hydrogen) oxidation in diabetic mice, and treatment of fenofibric acid, an active metabolite of fenofibrate, reduced mitochondrial oxidative stress and cell death in retinal neuronal cell lines under 4-hydroxynonenal (4-HNE)-induced oxidative stress conditions [98]. This implies mitochondrial dysfunction by oxidative stress could be restored by PPARα activation.

In terms of an ischemic model other than the diabetic model, PPARα^−/−^ mice with a laser-induced choroidal neovascularization (CNV) developed more severe CNV compared with wild-type CNV mice [99]. PPARα^−/−^ mice with oxygen-induced retinopathy (OIR) also showed deleterious effects (such as increased retinal cell death and glial activation) in comparison with wild-type OIR mice [100]. Overexpression of PPARα using an adenovirus system attenuated the increased endothelial progenitor cell circulation in OIR mice through the inhibition of the hypoxia-inducible factor (HIF)-1α pathway, and mouse brain endothelial cells from PPARα^−/−^ mice showed prominent activation of HIF-1α induced by hypoxia, compared with wild-type mouse brain endothelial cells [101]. This suggests a novel protective molecular mechanism of HIF-1α inhibition for anti-angiogenic effects of PPARα.

Pharmacological PPARα activation by palmitoylethanolamide (PEA) reduced retinal neovascularization and fibrotic changes and suppressed glial activation in proliferative retinopathy and neovascular age-related macular degeneration mouse models [102]. In cardiovascular studies, PEA exerted direct vaso-relaxation of the bovine ophthalmic artery through the PPARα transcription factors, suggesting a function of PPARα on physiological vascular regulation [103]. This vaso-relaxing effect could increase supply of oxygen to the retina and prevent ischemic lesions, as observed in patients with ocular hypertension [104]. Another study demonstrated that the administration of PEA showed enhancement of aqueous humor outflow facility and this effect appeared to be mediated partially by the involvement of PPARα [105]. Those studies imply PPARα modulation could be a promising therapeutic target for glaucoma [106].

Even though there are not many reports available on the roles of PPARα in the ocular surface, recent evidence suggests that PPARα may play a critical role in the regulation of inflammatory processes in the ocular surface [107]. Fenofibrate ameliorated the severity of ocular surface squamous metaplasia, commonly seen in patients with long-term deficiency of tear film [108] and suppressed the formation of tear film instability via the inhibition of macrophages and downregulation of pro-inflammatory factors [107]. In the corneal epithelium of mice with dry eyes by sleep deprivation, downregulation of PPARα expression was detected [109], and fenofibrate increased PPARα expression in cultured corneal epithelium sheets and restored microvilli morphology [109]. This implies PPARα activation could have potential for use as a preventive agent in patients with high risk of dry eye. Other cornea studies indicated therapeutic roles of PPARα agonists against corneal inflammation and neovascularization [110,111]. Corneal neovascularization is closely related to a reduction in corneal transparency which is important for visual acuity [112,113]. In rat corneal alkali burn models, corneal neovascularization was seen and a topical injection of fenofibrate suppressed its neovascularization through upregulation of PPARα mRNA expression and suppression of IL-6, IL-1β, Vegf and Ang-2 mRNA expressions [110,111].

The above, taken together as accumulating evidence, supports the concept that it may be important to restore or boost PPARα expression for the prevention of various ocular diseases. However, studies on the downstream signaling effector molecules regarding PPARα activation need to be further unraveled.

## 4. Functions of SPPARMα in the Eye

### 4.1. Effects of Pamafibrate in the Retina

Several studies have shown that pemafibrate has therapeutic effects on retinal diseases [114,115,116]. Our group showed that oral administration of pemafibrate, but not fenofibrate, increased plasma fibroblast growth factor 21 (FGF21) levels in OIR and inhibited retinal neovascularization through the inhibition of HIF-1α and *Vegfa* expressions [115]. In addition, long-acting FGF21 inhibited HIF activity in a photoreceptor cell line under hypoxic conditions [115]. Furthermore, we reported that oral administration of pemafibrate increased serum FGF21 levels in the streptozotocin-induced diabetic mouse model and preserved retinal function through maintaining the expression of synaptophysin which regulates synaptic vesicle endocytosis [114]. Additionally, long-acting FGF21 directly upregulated the expression of synaptophysin in differentiated neurons in vitro [114]. Interestingly, both studies have shown that pemafibrate did not increase the expression levels of FGF21 mRNA in the retina [114,115]. Therefore, pemafibrate could affect retinal diseases indirectly by upregulating systemic FGF21 levels to work on the damaged retina. In addition, another group showed that pemafibrate directly inhibited diabetes-induced vascular leukostasis and leakage in the rat retina through upregulation of THBD expression which encodes the glycoprotein thrombomodulin [116], and indicated that pemafibrate can increase the expression of thrombomodulin in human umbilical vein endothelial cells and human retinal microvascular endothelial cells in vitro [116]. Even though further studies are required to elucidate various mechanisms, they suggested that pemafibrate may directly affect endothelial cells. Thus, pemafibrate, a novel representative SPPARMα, could offer retinal protection through the upregulation of blood FGF21 levels or FGF21 expression in the liver to work on the damaged retina with suppression of pathological neovascularization [115], maintenance of retinal function [114], modulation of systemic metabolisms such as triglyceride or blood glucose levels [114]. Furthermore, pemafibrate could increase thrombomodulin expression in retinal endothelial cells to work on the damaged retina with suppression of vascular leakage, leukostasis and inflammation [116] (Figure 3). When it comes to the beneficial effects of pemafibrate compared with those of fenofibrate regarding diabetic retinal protection, pemafibrate may have a more considerable impact in respect of ameliorating DR than fenofibrate, although direct comparative studies are needed.

To date, a phase 3 randomized interventional trial, the pemafibrate to reduce cardiovascular outcomes by reducing triglycerides in patients with diabetes (PROMINENT) study, has been ongoing (ClinicalTrials.gov Identifier: NCT03071692). A total of 10,391 patients with dyslipidemia with type 2 diabetes were recruited for this study. The outcome measurement is associated with nonfatal myocardial infarction, nonfatal ischemic stroke, hospitalization for unstable angina or unplanned coronary revascularization, and cardiovascular death [117]. Pemafibrate has also been used in an investigation of the prevention of DR worsening in a clinical trial. However, this was terminated because the number of recruited subjects could not meet the criteria for the trial (PROMINENT-Eye Ancillary Study). Another phase 2 clinical trial for pemafibrate has been ongoing for nonalcoholic fatty liver disease (NAFLD) (ClinicalTrials.gov Identifier: NCT03350165). If pamafibrate has therapeutic effects for such metabolic syndromes, it could potentially influence retinopathies related to metabolic diseases, especially DR.

### 4.2. Effects of Fibroblast Growth Factor FGF21 on Retinopathy

FGF21 was discovered and first described in 2000 and now is known as a secreted protein composed of 209 amino acids [118]. FGF21 is a member of the FGF19 sub-family of signaling molecules. PPARs and PGC-1α (peroxisome proliferator-activated receptor γ coactivator 1-α) modulate FGF21 function, and a previous study showed that PPARα agonists ameliorated metabolic disorders in obese mice through modulation of FGF21 expression [119]. Several studies have shown that pemafibrate activates PPARα in human hepatocytes, resulting in an increase of FGF21 expression [81,120]. It has been reported that FGF21 had therapeutic effects on glucose and lipid metabolism in mice [121] and especially, long-acting FGF21 (PF-05231023) improved the levels of the circulating lipid profile in type 2 diabetic patients and in obese cynomolgus monkeys [122].

Several studies have shown that FGF21 has therapeutic effects on retinopathies in mice [123,124]. Fu et al. showed that PF-05231023 suppressed neovascularization in the OIR model, suppressing TNF-α expression through upregulating adiponectin expression [123]. They also detected mRNA expressions of FGFR1-4 and β-Klotho,crucial receptors for FGF21′s functions, in the total retina of mice. In addition, they indicated that long-acting FGF21 maintained retinal function in a streptozotocin-induced diabetic mouse model using electroretinography analyses [124]. Furthermore, we reported that long-acting FGF21 reduced retinal vascular leakage in vivo and in vitro via maintenance of claudin-1 expression [125]. In that study, we also detected expressions of FGFR1 and β-Klotho in human endothelial cells [125], which implies that the same strategy regarding the therapeutic role of FGF21 may also be applicable to humans. Retinal vascular leakage is seen in the early stage of DR patients [3,126]. Therefore, FGF21 administration or FGF21 induction by SPPARMα agonists can be a possible therapeutic for DR development at the early stage.

Pegbelfermin (BMS-986036), a polyethylene glycol-modified recombinant human FGF21 analog, has a prolonged half-life compared to other FGF21 drugs [127]. A phase 2B clinical trial has been ongoing to examine whether pegbelfermin has therapeutic effects on nonalcoholic steatohepatitis (NASH) and NAFLD (ClinicalTrials.gov Identifier: NCT03486899). It is desired that clinical studies for retinal diseases using this treatment be conducted in the near future. However, for the time being, this long-acting FGF21 should be subcutaneously-injected. Thus, we still need to wait for the development of oral treatment that will make administration easy for patients, which in turn may necessitate an alternative way to boost FGF21 levels in the body such as the administration of SPPARMα (pemafibrate).

## 5. Conclusions

In this review article, we summarized recent clinical studies of promising oral pharmacotherapies in DR and DME. In particular, we emphasized that PPARα agonists could possess the potential to protect against DR and described SPPARMα (pemafibrate) with its potential effects for various retinopathies, including DR.

Diabetes mellitus is a complex metabolic disorder which is associated with insulin resistance, insulin signaling impairment, β-cell dysfunction, abnormal glucose and lipid metabolisms, inflammation and mitochondrial oxidative stress [128]. Likewise, development of DR has multiple interlinked alterations via dysfunctions of various cell types with enormous complex pathological mechanisms. Its development in turn cannot be prevented or protected by one therapeutic molecular target. Fortunately, PPARα targeting could be an alternative therapy to the other therapeutic agents in that it covers systemic enhancement of glucose and lipid metabolisms, anti-inflammation, and anti-oxidative stress. Moreover, PPARα activation as well as FGF21, which acts on damaged retinas as an anti-neovascularization or a neuroprotective agent and is induced by SPPARMα, could ameliorate the development of DR at the early stage through prevention of retinal vascular leakage.

We hope that this review enables comprehensive understanding of protective roles of PPARα agonists against DR development. This review can be useful for future studies on the protective effects of PPARα agonists against DR.

## Figures and Tables

**Figure 1 biomedicines-08-00433-f001:**
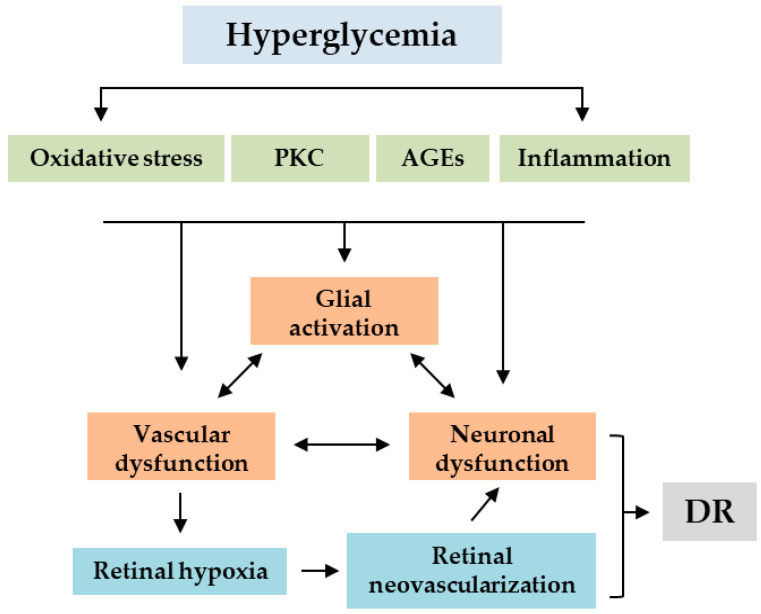
Schematic presentation of pathogenic mechanisms of diabetic retinopathy (DR). Hyperglycemia-induced metabolic stresses, such as mitochondrial oxidative stress, activation of protein kinase C (PKC) pathway, accumulation of advanced glycation end products (AGEs) and inflammation, induce impairment of crosstalk between neurons, glia and vasculatures through glial activation, vascular dysfunction and neuronal dysfunction. Retinal hypoxia induced by vascular dysfunction causes retinal neovascularization which worsens neuronal dysfunction in the retina, finally leading to DR. Double-headed arrows, interlinking process of events; single-headed arrows, one-way direction process of an event to the other event.

**Figure 2 biomedicines-08-00433-f002:**
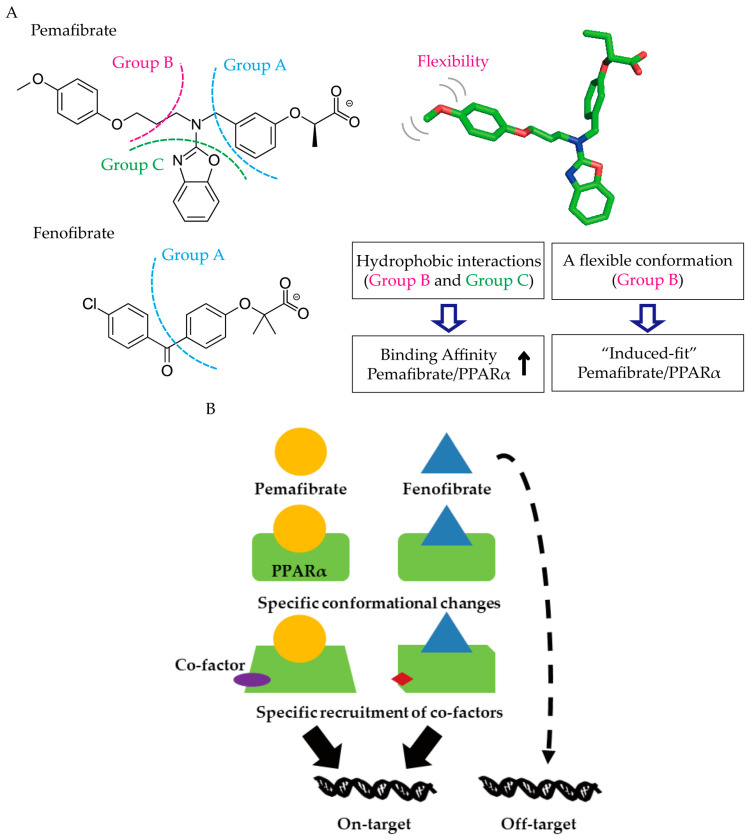
Structural differences between pemafibrate and fenofibrate. (**A**) The structure of pemafibrate contains the carboxylic acid group (Group A, blue color), phenoxy alkyl group (Group B, red color) and 2-aminobenzoxasole group (Group C, green color) while that of fenofibrate only contains Group A. This hydrophobic Y-structure of pemafibrate (Group B and C), which interacts with the hydrophobic residues in the ligand-binding pocket of PPARα, results in the improvement of fitting with the ligand-binding pocket of PPARα via increasing the receptor-ligand binding affinity. The flexibility of Group B confers the stronger “induced fit” conformation with PPARα. (**B**) The specific binding of pemafibrate can induce specific conformational transitions of PPARα with specific co-factor complexes resulting in exerting the on-target effects of PPARα activation while that of fenofibrate could exert the on-target as well as off-target effects other than PPARα activation such as deleterious effects on the renal function including increased serum creatinine levels. Solid lines: a direct series of an event; A dotted line: an indirect series of an event.

**Figure 3 biomedicines-08-00433-f003:**
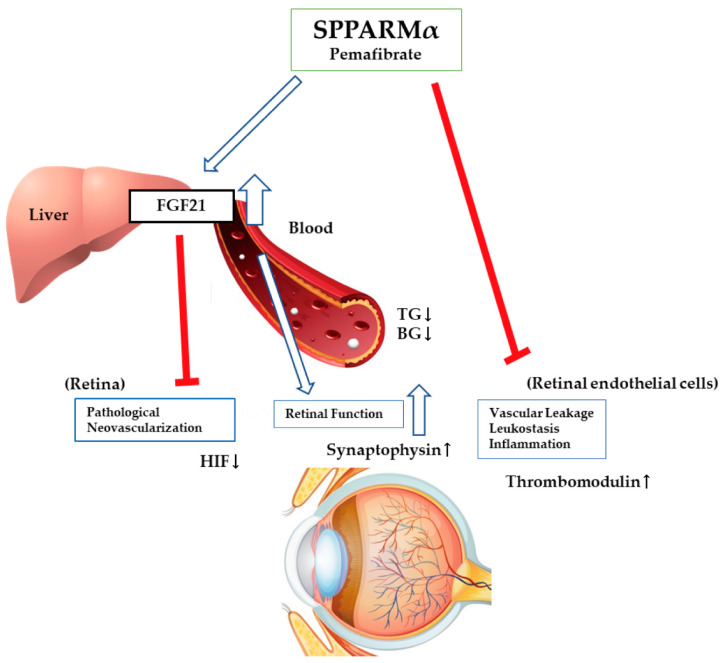
A schematic model of protective effects of a selective peroxisome proliferator-activated receptor alpha modulator (SPPARMα), pemafibrate, on injured retinas. Two therapeutic pathways are suggested. One is that pemafibrate upregulates fibroblast growth factor (FGF) 21 expression in the liver or blood and affects the retina or modulates abnormal metabolisms such as the reduction of increased serum triglyceride and blood glucose levels. The other way is pamafibrate directly affects retinal endothelial cells with vascular leakage, leukostasis, and inflammation. HIF; hypoxia-inducible factor, TG; triglyceride, BG; blood glucose. Inhibition of an event: red lines; Stimulation of an event: white lines with blue line borders.

**Table 1 biomedicines-08-00433-t001:** Recently investigated oral pharmacotherapies for DR or DME.

Therapeutic Agent	Molecular Target	Study Design (Subject/Treatment/Measurement)	Clinical Trial	Sponsor or Collaborator
α-lipoic acid	Antioxidant	Diabetes type II subjects, mild non-proliferative DR	Phase III	Ludwig-Maximilians—University of Munich/Bausch & Lomb Incorporated
		α-lipoic acid (600 mg per day) vs placebo		
		Occurrence of the clinically significant macular edema		
Ubiquinone	Antioxidant	Diabetes type II subjects, non-proliferative DR	Phase II	University of Guadalajara/Instituto Mexicano del Seguro Social
		Ubiquinone (400 mg per day) vs. placebo		
		Alteration in the activities of oxidative stress markers		
Aminoguanidine	AGEs inhibitor	Diabetes type I subjects, DR	Phase I	University of Minnesota
		Aminoguanidine (150 mg 1.5 h before measurements)		
		Alteration in the vascular response to flicker		
ASP8232	VAP-1 inhibitor	Diabetes type I or II subjects, DME	Phase II	Astellas Pharma Europe B.V.
		ASP8232 (an capsule per day) vs placebo, w/ ranibizumab		
		Change in the central retinal thickness		
BI 1467335	VAP-1 inhibitor	Diabetes type I or II subjects, non-proliferative DR	Phase II	Boehringer Ingelheim
		BI 1467335 (once per day) vs placebo		
		Change in any ocular events according to Common Terminology Criteria for Adverse Events		
Minocycline	Microglial activation inhibitor	Diabetes type I or II subjects, DME	Phase I, II	National Eye Institute/The Emmes Company, LLC
		Minocycline (100 mg twice per day)		
		Alteration in the visual acuity and retinal thickness		
Fenofibrate	PPARα agonist	Diabetes type I or II subjects, DR	Phase IV	University of Padova/Azienda Ospedaliera di Padova
		Fenofibrate (145 mg per day) vs placebo		
		Alteration in endothelial progenitor and circulating progenitor cell levels		
SLV348	PPARα agonist	Diabetes type II subjects, DME	Phase II	Abbott Products
		SLV348 (135 mg per day) vs placebo		
		Alteration in total macular volume		

DR, Diabetic Retinopathy; DME, Diabetic Macular Edema; PPARα, peroxisome proliferator-activator receptor alpha.

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
