# Peer review of "PPARα Agonist Oral Therapy in Diabetic Retinopathy"

_biomedicines, 2020, doi:10.3390/biomedicines8100433_

Round 1

Reviewer 1 Report

The authors reviewed the PPARα agonists with potential applicability in an early stage of Diabetic Retinopathy. Although the manuscript has some potential, the structure of the manuscript is very confusing and the manuscript writing should be carefully reviewed since it becomes very difficult to read and understand it.

The manuscript should be carefully reviewed. Some grammatical errors and incorrect spelling make the manuscript very difficult to read and understand.

Some examples:

Abstract – “Diabetic retinopathy (DR) is an eye condition, develops after chronically poorly-managed diabetes, and is presently the main cause for blindness on a global scale.” should read “… is an eye condition that develops …cause of blindness”

Abstract – “Moreover, the ways of the treatments are invasive to the eyes. Therefore, new pharmacological treatments for the early stages of DR and the safe and effective ways of the treatments still remain needed.” Should read “Moreover, the forms of treatment available for DR are highly invasive. … safer and more effective pharmacological treatments are required for DR treatment, in particular in an early stage.” (The same applies to the lines 50-53)

Line 138 – “PPARs are nuclear receptors which regulate several gene expressions affecting lipid and carbohydrate metabolism [51].” should be  “PPARs are nuclear receptors that regulate the expression of several genes affecting lipid and carbohydrate metabolism [51].”

Lines 37-42 - “In detail, DR progresses from the stage that at first high blood glucose level damages the 38 microvasculature [3]. Then, the microvascular irregularities including hemorrhage, ischemia, and microaneurysms bring retinal neovascularization [3]. Abnormal vasculatures by retinal neovascularization lead to a severe hypoxic condition in the retina of the eye [3]. At the final stage, fibrovascular proliferation resulting in tractional retinal detachment by chronic severe hypoxic conditions causes vision loss [3].” I miss the reference of other pathological events in diabetic retinopathy (DR) in this paragraph, including inflammation and neuroglial degeneration. As matter of fact, recent studies indicate that neurodegeneration may precede microvascular changes that occur in diabetic retinopathy. It will be relevant to further reinforce the importance of PPARα agonists could have in DR since it attenuates inflammation (referred in the manuscript) but also that it can be a relevant target for neuroprotection (barely referred in the manuscript (lines 223-226)). Furthermore, this fact gains more relevance since the authors refer that PPARα agonists induce the expression of FGF, which is considered a neurotrophic factor secreted in the retina by cells such as Müller cells.

Lines 63-64 – “Recently investigated oral pharmacotherapies in clinical trials for DR or diabetic macular edema are listed in Table 1 (https://clinicaltrials.gov) (last updated, 06 Sep 2020).”  How the authors selected the more relevant oral therapies for DR and diabetic macular edema shown in table 1?

Lines 68 – 136 – I understand the interest in referring to other oral pharmacotherapies that are currently being studied for DR. However, I hoped the authors do some comparison between these, including their disadvantages and potential side effects in comparison PPARα agonists (that is the main goal of the manuscript). It is not clear that PPARα agonists are advantageous compared to the other oral pharmacotherapies.

Lines 140-141 “Among them, PPARa has been spotlighted in that the expression levels have been shown to be reduced in the retinas with diabetes [53].” and lines 195-197 “ Decreased PPARa expression in the rat diabetic retinas was considerably observed compared with that in the non-diabetic retinas [79].” The authors should remove repeated and redundant information.

Line 20 - “treatments for the early stages of DR” and Lines 22-25 - Especially, we focus on peroxisome proliferator-activator receptor alpha (PPARα) agonists in that experimental evidence suggests PPARα activation is closely related to the attenuation of vascular damage, including lipid-induced toxicity, inflammation, an excess of free radical generation, endothelial dysfunction and angiogenesis. The abstract promises more information than the one is available in the manuscript. Indeed, the effects described are more focused on the effects in the proliferative phase of DR (e.g. inhibition of VEGF and neovascularization). However, I expected that the manuscript would describe the potential application of PPARα agonists in an early phase of DR, in which more efficient therapies are required.

Reviewer 2 Report

This is a succinct but well -rounded review on therapies against diabetic retinopathy, especially focusing on PPAR-αagonists. The authors have referenced the latest papers  in the foiled. I have a couple of minor points:

only :

  1. A bit more detailed introduction to diabetic retinopathy, on the pathogenesis, at the beginning would be better for readers who are new to the field.
  2. Line 137: It would help readers not familiar with PPAR-α-  a brief introduction to the protein
  3. Line 160: “more hydrophilic” Should this be “more hydrophobic”?

Reviewer 3 Report

In this manuscript, the authors aim to summarize the recent clinical studies of potential oral pharmacotherapies in DR, especially PPARα agonist. The topic on using PPARα agonist to manage DR has not been reviewed for quite some time and is therefore a good topic for a review. However, this manuscript is very poorly structured and lack of depth and comprehensiveness. It requires very major restructuring and rewriting before I can consider recommending it for publication.

  1. In the Introduction section, apart from briefly explaining diabetic retinopathy (DR), the authors should also briefly explain (about one paragraph) the functions of PPARα in the body.
  2. In Section 2 on An Overview of Oral Therapies, the authors used about two pages explaining Antioxidants, Advanced Glycation End Products (AGEs) Inhibitor, Vascular Adhesion Protein-1 (VAP-1) Inhibitors and Microglial Activation Inhibitor in treating DR. As a review focusing on “PPARα Agonist Oral Therapy in Diabetic Retinopathy”, using two pages (the main text is only about 8-9 pages!) explaining oral therapies unrelated to PPAR in treating DR is totally unnecessary. If the authors intend to write a comprehensive review on all oral therapies used in treating DR, then these oral therapies should also be discussed in detail in the later sections (and 8-9 pages of main text is definitely insufficient for this review).
  3. For Section 3 on Functions of PPARα in the Eye, this section should be one of the important sections for this review and should be written in depth. Yet, the authors only used one paragraph to summarize some research findings in this area. I recommend the authors to read some previous reviews on the functions of PPARα in the eye and rewrite this section systematically and in detail.
  4. Section 4 on Functions of SPPARMα in the Eye is also written very briefly. Please read other previous reviews in this area to restructure your review sections.
  5. Figures are okay.
  6. Although there is almost no grammar mistake in the manuscript, some sentences are too long and difficult to comprehend. Sentences within some paragraphs also does not flow logically and should be reorganized.

In summary, the topic chosen by the authors is good and worth reviewing. However, my major criticism for this manuscript is the lack of detail and depth, especially in the very important sections. The review manuscript definitely does not meet the goal to “enable more comprehensive approaches for understanding protective roles of PPARα in the prevention of DR development” as mentioned in the abstract. Much more effort should be put into this manuscript by the authors to make it comprehensive and impactful.

Round 2

Reviewer 1 Report

The authors reviewed the PPARα agonists with potential applicability in an early stage of Diabetic Retinopathy.  The manuscript was carefully reviewed and, therefore, its quality has greatly improved. The authors made a very good job of improving sections 1. Introduction (I particularly enjoy the inclusion of figure 1 and associated description) and 3. Functions of PPARα in the Eye.  

My only concern about the revised manuscript is section 2. An Overview of Oral Therapies. Considering that is not enough information to compare adequately the several oral drugs understudy and PPARα agonists, as recommended, I still consider that the other oral drugs have the same highlight that the PPARα agonists (that are the focus of the review).

Reviewer 3 Report

The authors have made significant changes to the manuscript to address the reviewers' comments and I thank them for the effort. The sections have improved significantly in quality. However, I still have one last concern regarding this manuscript before acceptance.

1. The authors have decided to keep the section "An Overview of Oral Therapies" for this manuscript. I still feel that this section is out of place for the goal of this manuscript. Hence, I suggest that in order for the authors to keep this section, the authors should rename this section as "Comparison of PPARα Agonist and Other Oral Therapies". After renaming the section, the author can explain in detail the other oral therapies since it is now a section for other oral therapies to compare with PPARα Agonist oral therapies and this will not compromise the aim of this manuscript.
